# The relationship between loneliness and mobile phone addiction among Chinese college students: The mediating role of anthropomorphism and moderating role of family support

**Yanhong Zhang**[1☯], **Yongxue Li**[2☯], **Meng Xia**[1], **Miao Han**[1], **Lei Yan**[1], **Shuailei Lian** [1]*

1 Department of Psychology, College of Education and Sports Science, Yangtze University, Jingzhou, China,
2 Faculty of Psychology, Southwest University, Chongqing, China

☯ These authors contributed equally to this work.

* lslyzpsy@yangtzeu.edu.cn

**Data Availability Statement:** All relevant data are within the manuscript and its Supporting Information files.

## Abstract

### Background

Loneliness and mobile phone addiction are common phenomena in our daily life that can lead to physical and psychological maladaptation. Although loneliness has been considered to be correlated with adolescents' mobile phone addiction, the underlying mechanisms of the relation between loneliness and mobile phone addiction are still unclear. In order to address the problem of mobile phone addiction among adolescents, the association of loneliness with mobile phone addiction was explored from the perspective of Compensatory Internet Use Theory. The mediating role of anthropomorphism and the moderating role of family support were examined.

### Methods

Data were collected through convenience sampling from a comprehensive university in China. A total of 582 Chinese college students (279 men and 303 women), aged from 17 to 24 ($M_{age} = 20.22$ years, $SD = 1.46$), participated in this study. Their loneliness, anthropomorphism, family support and mobile phone addiction were measured using well-validated self-report questionnaires. Regression-based statistical mediation and moderation were conducted using the PROCESS macro for SPSS.

### Results

Loneliness was significantly and positively associated with mobile phone addiction, and this link could be mediated by anthropomorphism. Moreover, the direct effect of loneliness on mobile phone addiction and the indirect effect of anthropomorphism in this link were moderated by family support. Both these two effects were stronger for college students with lower family support.

**Funding:** This work was supported by the Major Research Project of Philosophy and Social Science in Hubei Province (Project No. 21ZD045).

## Conclusion

The present study broadened our knowledge of the underlying mechanisms between loneliness and mobile phone addiction. These findings may contribute to a better understanding of the impact loneliness can have on mobile phone addiction as well as its paths. Moreover, the results can also provide advice for parents and constructive suggestions for rationalizing college students' mobile phone use in the mobile Internet era. Educational professionals and parents should pay special attention to the problem of mobile phone addiction among lonely adolescents, especially for those with low family support.

## Introduction

With the progressive expansion of digital technology, mobile phones, especially smartphones, play an important role in people's daily lives, and the proportion of users has increased [1]. According to the 50th Statistical Report on Internet Development in China, the number of Internet users in China reached 1.051 billion by June 2022. Among them, mobile phone users accounted for 99.6% [2]. It is worth noting that use of mobile phones is a double-edged sword, which brings people great convenience on the one hand, but causes psychological problems on the other, such as mobile phone addiction, pain and weakness in the thumbs and wrists [3], poor sleep quality [4] and high rates of depression [5]. Since the arrival of smartphone devices, mobile phones quickly became a source of potentially addictive behavior [6]. As a kind of technological addiction [7], mobile phone addiction, sometimes termed problematic mobile phone use, excessive smartphone use, smartphone addiction and mobile phone dependence, refers to a fundamentally similar concept of mobile phone use as a compulsive and excessive usage of mobile phones [6,8]. Despite various labels, mobile phone addiction has been operationalized fairly consistently as a multidimensional construct. The key dimensions include four common factors: (a) the use of mobile phones is out of control; (b) obsessive thoughts about mobile phones (craving); (c) experiencing anxiety and other negative emotions when unable to use mobile phones (withdrawal); and (d) despite rising economic costs, more and more time is spent on mobile phones (tolerance) [9,10]. Mobile phone addiction has become a widespread concern of researchers. Thus, exploring the factors and underlying mechanisms of mobile phone addiction are of great practical value for developing interventions targeting people with mobile phone addiction and promoting the value of maintaining a healthy life.

In recent years, researchers have conducted in-depth research into mobile phone addiction to better prevent and control it. Many studies have explored the risk factors closely related to mobile phone addiction, among which loneliness is one of the most concerning [11]. Loneliness is defined as a subjective psychological state caused by the failure of interpersonal relationships to reach expected levels, often accompanied by boredom, helplessness, depression, and other negative psychological experiences [12,13]. Potential sources of loneliness include people's various relationships (e.g., peer-related loneliness and parent-related loneliness [14]). College students facing many developmental tasks, such as separation from parents and establishing relationships with peers [15], might be particularly subject to loneliness [16]. According to the compensatory Internet use theory (CIUT), the Internet is utilized to alleviate negative feelings which increase a motivation to go online [17]. As a portable surfing tool, a mobile phone is easy for people to acquire and go online. The accessibility and multi-functionality of mobile phones help individuals to establish virtual online social relations to satisfy their sense of belonging and can also relieve their helplessness, loneliness and other negative emotions

through online entertainment [12,18]. When an individual with a high level of loneliness initially tries to use a mobile phone, such behavior is possibly reinforced by removal or reduction of negative emotion existing previously [19]. The process will drive people to continue to use a mobile phone and then develop mobile phone addiction. Meanwhile, research focusing on the direct link between loneliness and mobile phone addiction has also illustrated that individuals suffering from loneliness are more inclined to become mobile phone addicts. In particular, studies have shown that the higher the loneliness level, the more likely individuals are to use mobile phones [20]. Therefore, we proposed the following hypothesis:

Hypothesis 1: Loneliness would positively associated with mobile phone addiction among Chinese college students.

Given the negative consequences of smartphone addiction, examining the underlying mechanisms involved in mobile phone addiction is necessary. Research have conformed the effects of motivations on psychopathological disorders and addictive symptoms of the internet or smartphones usage [21,22]. The Cognitive-behavioral Model (CBM) considers negative self-perception can occur when people are (a)motivated to use the internet or have psychosocial disorders. these distal and proximal factors reinforce the development of addiction symptoms [23]. According to the three-factor theory of anthropomorphism, elicited agent knowledge works as a cognitive mechanism, whereas sociality and effectible are motivational mechanisms. Anthropomorphic knowledge and knowledge concerning humans is activated and applied when effectible and social motivation is increasing [24,25]. Therefore, the study would explore that loneliness can perform as the psychosocial factor influencing mobile phone addiction, and anthropomorphism might serve as a mediating cognition variable, which leads to mobile phone addiction. Furthermore, family support has proven to be an effective buffer against the adverse effects of risk factors on individuals' problem behaviors [26]. Thus, family support is examined as a moderator variable. In sum, the present study investigated whether anthropomorphism mediated the relation between loneliness and mobile phone addiction and whether this direct and indirect path was moderated by family support.

## The mediating role of anthropomorphism

The term of anthropomorphism is defined as the psychological process or individual difference of imbuing non-human objects with human-like characteristics, motivations, intentions, or mental states, believing inanimate objects as having human qualities [27]. According to the three-factor theory of anthropomorphism, anthropomorphism results from chronically unfulfilled social connections [27]. People as social animals, are in various social networks from birth. These connections with others are the key resources for human survival, reproduction, and even development. Social motivation is the need for people to seek social contact, social connection and social recognition. If this need cannot be satisfied, people might use an alternative strategy: Nonhuman entities, to meet this basic need by anthropomorphizing a connection with nonhuman subjects [27,28]. However, when people perceive that their interpersonal communication is not up to the ideal expectation, they will have the feeling of loneliness [29]. Individuals that suffer from loneliness have a strong need to re-establish social relationships [30]. Loneliness not only makes individuals yearn for social relations more [31], but also urges individuals to make up for the lack of interpersonal relations of various ways, so as to regain the sense of belonging and alleviate the negative experience brought by loneliness. Anthropomorphizing objects can provide convenience for individuals suffering from loneliness to form effective social and emotional connections with objects [32]. Because of the emergence of anthropomorphism provides a possibility to satisfy the need to connect with people and to maintain a sense of belonging [31]. Additionally, empirical studies on the direct link between

loneliness and anthropomorphism also verified that loneliness was an important predictor [33]. Therefore, we conducted that loneliness is plausibly positively associated with anthropomorphism.

As for mobile phone addiction, though the current studies have not directly examined its relationship with anthropomorphism, relevant studies can provide indirect evidence. Previous studies have shown that anthropomorphism is an important predictor of various problematic behaviors [34,35]. For instance, Neave et al. found that anthropomorphism was significantly associated with excessive object attachment [35]. Hazan and Shaver based on Bowlby's attachment theory, claimed that people usually become attached to multiple individuals, and even to inanimate objects in some cases, such as smartphones [36]. In the age of the Internet, more and more people find it difficult to get rid of their mobile phones, and this kind of phenomenon gives mobile phones special emotional function, leading to mobile phones gradually becoming objects of human anthropomorphism and attachment [37]. Anthropomorphism, as a negative cognitive bias and immaturity performance [38], gives nonhuman objects human-like characteristics and alters people's emotional and cognitive responses to the objects [39]. Mobile phones are considered as "companion", "secretary", "friend", or "safety ward" [40]. Liu and Ji found a significant positive correlation between mobile phone anthropomorphism and mobile phone affectional attachment [41]. Therefore, when individuals' psychological needs cannot be satisfied, they will seek comfort from the cell phone as an attachment object, and when individuals will over dependent on it, addiction will eventually result. Moreover, the cognitive-behavioral model of pathological Internet uses (PIU) supposes that of the chain of factors leading to a range of behavioral symptoms, some factors are located in the proximal chain whereas others are located in the distal chain away from the range of behavioral symptoms [42]. Loneliness is a distal cause of mobile phone addiction and an essential etiological element. Maladaptive cognitions such as anthropomorphism are proximal sufficient causes of mobile phone addiction, as they are sufficient to cause the set of symptoms associated with mobile phone addiction, such as excessive usage of mobile phones. Thus, maladaptive cognitions such as anthropomorphism plausible serve in a mediating role for the association between loneliness and mobile phone addiction. That is, loneliness leads people to engage in anthropomorphism which in turn facilitates mobile phone addiction. Consequently, we proposed the following hypothesis:

Hypothesis 2: Anthropomorphism may act as a mediator in the link between loneliness and mobile phone addiction.

## The moderating role of family support

As previously stated, exploring the mediating role of anthropomorphism in the relationship between loneliness and mobile phone addiction can answer the question of how does loneliness affect mobile phone addiction, but it cannot clarify when the effect is more likely. According to the perspective in theory of ecological techno-subsystem [43], the relationship between individuals and their mobile phone should be integrated into the social environment. As an important external environment factor for people's development, the material, emotional and social support provided by the family can provide a useful buffer or protection for individuals under the stress of loneliness [44].

Family support refers to the ability and willingness of families to support each other during difficult times [45] and it has been found to be a positive predictor for maintaining individuals' mental health and adaptation [46]. Numerous empirical studies have confirmed that positive parenting, such as a high level of love and support, positive communication, parent-child attachment, and parental monitoring, are beneficial in promoting individuals' healthy development [47], and in playing an important role in preventing and intervening in mobile phone

addiction [20]. On the contrary, parental rejection and neglect are not conducive to the healthy development of individual's physical and mental health and can cause bad behaviors (e.g., mobile phone addiction, Internet addiction [48]). Under the deficient self-regulation perspective, individuals with psychosocial problems will have deficient self-regulation and control ability, which contribute to the amount of time one spends using the mobile phone, and eventually evolves into mobile phone addiction [49]. People with high levels of loneliness tend to have poor self-regulation and have difficulty maintaining healthy media use habits, making them more likely to become addicted to their phones than individuals with low levels of loneliness [13]. As an environmental variable, family support positively moderates individuals' negative emotions [50]. The lack of perceived parental support, combined with feelings of loneliness, might expose people to inner unpleasant feelings, consequently reinforcing compulsive-impulsive behavior as a maladaptive coping strategy [16]. Family support can effectively alleviate the adverse psychological reactions caused by negative emotions, so as to ensure that individuals are at a healthy level of body and mind [50]. Meantime, high levels of parental behavior monitoring are beneficial for individuals to form higher self-control ability [51]. The resilience framework theory points out that resilience, including family support, is an important protective factor, that plays a moderating role in risk factors and individual development, reducing the negative consequences produced by risk factors and promoting the psychological adaptation and development of individuals [52]. Empirical studies have also shown that social support can mitigate the effects of negative emotion on mobile phone addiction and problematic mobile phone use [53]. Therefore, family support might moderate the adverse effects of loneliness on mobile phone addiction. That is, loneliness would lead to addiction largely when family support is low rather than when it is high.

The physical, emotional, informational, and instrumental assistance that individuals perceive from their family members often serves as an external coping resource for negative symptoms [54]. According to the three-factor theory of anthropomorphism [27], people's need to establish and maintain a sense of social interaction with others, including non-human subjects. As a resource to cope with stress, family support can effectively mitigate the negative effects of risks by mobilizing an individual's positive emotions and creating supportive social networks [55]. For instance, a positive and warm family atmosphere and close connection to family could provide important support for people dealing with negative emotions (i.e., loneliness) by increasing the sense of belonging, security, and self-worth [56]. On the contrary, negative parental emotional expressivity can cause tension in the family emotional atmosphere and threaten an individual's sense of attachment [57]. In addition, the risk-buffering model indicates that positive factors in the environment can reduce the adverse effects of risk factors [58]. High family support, as a positive factor in the living environment, can play a protective role when individuals experience some negative events [59]. Individuals with high levels of family support have more social support and establish secure attachments, thus mitigating the effects of negative emotions [60], that is to reduce the negative impact on loneliness on anthropomorphism. Therefore, family support is predicted to moderate the adverse effects of loneliness risk factors on anthropomorphism.

Above all, we propose the hypothesis as follows:

Hypothesis 3: Family support would moderate the relationship between loneliness and anthropomorphism as well as mobile phone addiction, with the relationship being weaker for individuals with high family support.

## The current study

Based on the literature review and research hypothesis, a moderated mediation model was constructed in current study. This study further examined the relationship of loneliness,

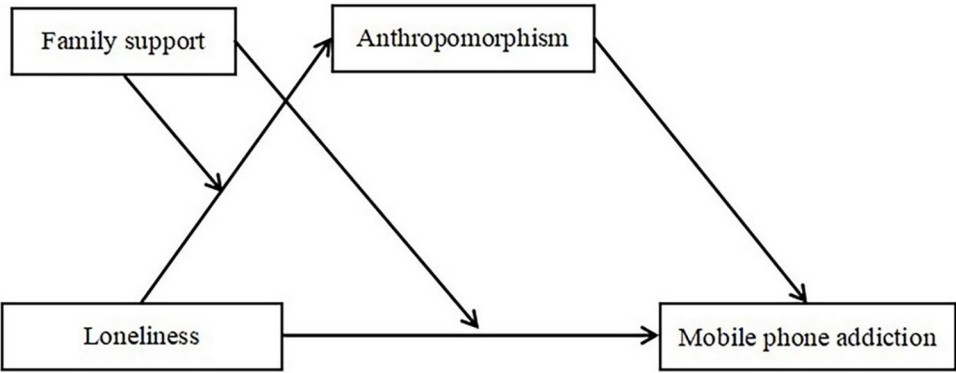

**Fig 1. The proposed moderated mediation model.**

anthropomorphism, family support, and mobile phone addiction by assuming a comprehensive model. Overall, this study answers two main questions: First, it was hypothesized that anthropomorphism would mediate the relationship between loneliness and mobile phone addiction. Second, family support would moderate the direct and indirect association between loneliness and mobile phone addiction. Specifically, how does loneliness contribute to mobile phone addiction, and when is this association most potent? The proposed model is illustrated in Fig 1. The model can help us shed light on the mediation and moderation mechanisms underlying the relation between loneliness and mobile phone addiction.

## Materials and methods

### Participants and procedure

In this study, we used G*Power3.1.9.7 to calculate the sample size [61], the calculated parameters including Tails = two, Effect size = 0.2, α err prob = 0.05, Power (1-β) = 0.95, calculated sample size is 314. Considering the invalid response rate of the subjects, assuming that the invalid response rate is 20%, 314 / (1–0.2) = 393 questionnaires should be sent out. Participants were recruited from a Web site (https://www.wjx.cn/) in China with the approval of the school authority and the Academic Committee for Scientific Research at the corresponding author's university. Data collection was performed during week 9, 2021 (March 1–7). Inclusion criteria for this study were full-time college students, filling in longer than 3 mins, and volunteering to participate in this survey. Participants were informed of the requirements of the survey by using common instructions, emphasizing the authenticity, independence, and integrity of all answers. All the participants agreed to answer the questionnaire. After giving informed consent, participants completed demographic and self-report questionnaires (see the Questionnaires section). We collected a total of 624 participants' data, but we excluded 42 participants according to the completion time (i.e., they completed in less than three minutes), and finally got 582 valid data, with an effective rate of 93.27%. The final sample consisted of 279 men and 303 women participants aged between 17 and 24 ($M_{age}$ = 20.22 ± 1.46). One hundred and twenty-two (20.96%) of them were freshmen; one hundred and forty-two (24.40%) of them were sophomores; one hundred and seventy-nine (30.76%) of them were juniors; One hundred and thirty-nine (23.88%) of them were seniors.

### Measurement

**Loneliness.**    The Chinese version [62] of the UCLA Loneliness Scale was adopted to assess participants' loneliness [63]. The scale consists of 20 items rated on a 4-point Likert scale

(1 = never, 4 = always). A sample of an item is "Do you often feel lonely?" All the items were averaged after 9 items were reverse-scored. Overall, higher scores indicated higher level of loneliness. Cronbach's alpha for the scale in the current study was 0.83.

**Mobile phone addiction.** As has been widely used in a diversity of samples to measure individuals' mobile phone addiction tendency, the Mobile Phone Addiction Index (MPAI) was used in this study [8]. The scale has been proven to have good reliability and validity for Chinese students and young adults [4]. This scale consists of seventeen items (e.g., "You have attempted to spend less time on your mobile phone but are unable to"). Participants responded on a Likert-type scale ranging from 1 (never) to 5 (always). Responses were averaged to form an overall measure of students' mobile phone addiction, with higher scores indicating greater mobile phone addiction. The items also demonstrated high reliability in the present study (Cronbach's $\alpha$ = 0.92).

**Anthropomorphism.** The Anthropomorphism Questionnaire (AQ) was adopted to measure anthropomorphism [34]. This scale includes two subscales: child ($AQ_{Child}$) and current ($AQ_{Current}$), each subscale has ten items (e.g., "When I was a child, I made sure that when I put my toys away the ones who were friends were placed side by side", "On occasions I feel that my computer/printer is being deliberately awkward"). Although the questionnaire has two subscales, they are typically highly correlated and the current study just uses the aggregate score. Participants were asked to respond on a 7-point Likert scale ranging from 0 (not at all) to 6 (very much so). Higher scores represent higher anthropomorphism. Cronbach's alpha for the scale in the current study was 0.96.

**Family support.** The family support subscale in the Chinese version of the Adolescent Resilience Scale [64] was adopted to assess participants' family support. The subscale consists of six items (e.g., "Parents respect my opinion very much"). Participants responded on a Likert-type scale ranging from 1 (strongly disagree) to 5 (strongly agree). Higher scores represent higher perceived family support. Cronbach's alpha for the subscale in the current study was 0.65.

**Statistical analysis.** In this study, a moderated mediation model was constructed with loneliness as the independent variable, mobile phone addiction as the dependent variable, anthropomorphism as the mediating variable, and family support as the moderating variable. Statistical analyses were conducted using SPSS 24.0. Since self-report data were collected for the present study, Harman single factor test was conducted to test the potential common method biases before data processing [65]. A total of 63 items of four main variables were tested. The results showed that there were eleven distinct factors with eigenvalues greater than one. The maximum explaining rate of the first factor was 24.67%, which is below the threshold level of 40% [66]. Therefore, common method bias was not obvious in the present study.

After common method bias evaluation, we carried out the following data processing steps. Firstly, we employed descriptive statistics and Pearson correlation analysis to examine the means, standard deviations, and bivariate associations of the study variables. Secondly, sobel test was employed to examine the predicted indirect effect of anthropomorphism in the relationship between loneliness and mobile phone addiction. The absolute value of z-value greater than 1.96 indicates that the mediating effect is significant. Thirdly, the SPSS macro PROCESS (Model 8) suggested by Hayes ([67], http://www.afhayes.com) was used to test the moderated mediation model. This SPSS macro has been used to test mediating and moderating models in several studies, in which this SPSS macro showed higher statistical testability [4,68,69]. The proposed moderated mediation model, name as Model 8, tests the influence of a moderator on a mediation model, with the moderation occurring on the direct path (the relationship between the independent variable and dependent variable) and the first half of the indirect path (the relationship between the independent variable and the moderator) of the mediation

model. Specifically, we used Model 8 to test the moderating effect of family support on the direct relationship between loneliness and mobile phone addiction, and the indirect relationship through anthropomorphism (the link between loneliness and anthropomorphism). Bootstrap confidence intervals (CIs) were applied to determine whether the effects in Model 8 was significant from 5,000 random samples of the data. CIs excluding zero indicated significant effects. Furthermore, simple slopes analyses were performed to decompose all the potential significant interaction effects [70]. Additionally, gender, age as well as grade were included as control variables in our model, as previous studies found that they were mostly related to the main variables in this study [11,48,53].

## Results

### Preliminary analyses

The descriptive statistics and correlation matrix for the study variables are presented in Table 1. Correlation analyses indicated that loneliness was negatively correlated with family support ($r$ = -0.39, $p <$ 0.01) and positively correlated with anthropomorphism ($r$ = 0.12, $p <$ 0.01) and mobile phone addiction ($r$ = 0.17, $p <$ 0.01). Anthropomorphism was positively correlated with mobile phone addiction ($r$ = 0.32, $p <$ 0.01) and negatively correlated with family support ($r$ = -0.08, $p <$ 0.05). Mobile phone addiction was not significantly associated with family support ($r$ = -0.05, $p >$ 0.05).

### Testing for the proposed moderated mediation model

Hayes's [67] SPSS macro PROCESS was adopted to examine the proposed moderated mediation model. Table 2 presented the main results.

As expected, the mediator variable model ($F_{(6, 575)}$ = 2.49, $R^2$ = 0.03, $p <$ 0.05) and dependent variable model ($F_{(7, 574)}$ = 13.23, $R^2$ = 0.16, $p <$ 0.001) were all significant after controlling for gender, age and grade. In specific, loneliness positively predicted anthropomorphism ($B$ = 0.38, $p <$ 0.05) and mobile phone addiction ($B$ = 0.28, $p <$ 0.001). Anthropomorphism positively predicted mobile phone addiction ($B$ = 0.17, $p <$ 0.001). Furthermore, Sobel test was employed to examine the significance of the indirect effect of loneliness on mobile phone addiction via anthropomorphism. The results indicated that anthropomorphism significantly mediated the relationship between loneliness and mobile phone addiction ($z$ = 2.31, $p <$ 0.05). These results provided compelling evidence that loneliness was associated with increasing in

**Table 1. Descriptive statistics and interrelations among all of the observed variables.**

| Variables | M | SD | 1 | 2 | 3 | 4 | 5 | 6 | 7 |
|---|---|---|---|---|---|---|---|---|---|
| 1.age | 20.22 | 1.46 | 1 | | | | | | |
| 2.grade | – | – | 0.78** | 1 | | | | | |
| 3.gender | – | – | -0.22** | -0.18** | 1 | | | | |
| 4.Loneliness | 2.24 | 0.42 | -0.03 | -0.02 | -0.03 | 1 | | | |
| 5.Anthropomorphism | 2.31 | 1.44 | -0.06 | -0.03 | -0.01 | 0.12** | 1 | | |
| 6.Mobile phone addiction | 2.91 | 0.80 | -0.04 | -0.02 | 0.15** | 0.17** | 0.32** | 1 | |
| 7.Family support | 3.40 | 0.67 | -0.07 | -0.08* | 0.11** | -0.39** | -0.08* | -0.05 | 1 |

Note. $N$ = 582.

**$p <$ 0.01

*$p <$ 0.05.

**Table 2. Regression results for the mediated moderating effect.**

| Model | | | | | | |
|---|---|---|---|---|---|---|
| **Model 1 (Outcome: anthropomorphism)** | | | | | | |
| *R* | *R²* | *F* | *B* | *Boot* LLCI | *Boot* ULCI | *t* |
| 0.16 | 0.03 | 2.49* | | | | |
| Gender | | | -0.05 | -0.30 | 0.20 | -0.37 |
| Age | | | -0.07 | -0.20 | 0.06 | -1.07 |
| Grade | | | 0.03 | -0.14 | 0.19 | 0.32 |
| Loneliness | | | 0.38* | 0.03 | 0.68 | 2.48* |
| Family support | | | -0.09 | -0.28 | 0.10 | -0.92 |
| Loneliness × Family support | | | -0.34* | -0.65 | -0.02 | -2.13* |
| **Model 2 (Outcome: mobile phone addiction)** | | | | | | |
| *R* | *R²* | *F* | *B* | *Boot* LLCI | *Boot* ULCI | *t* |
| 0.40 | 0.16 | 13.23*** | | | | |
| Gender | | | 0.24*** | 0.11 | 0.36 | 3.71*** |
| Age | | | 0.01 | -0.06 | 0.07 | 0.18 |
| Grade | | | 0.01 | -0.08 | 0.10 | 0.24 |
| Loneliness | | | 0.28*** | 0.13 | 0.44 | 3.51*** |
| Anthropomorphism | | | 0.17*** | 0.11 | 0.22 | 6.04*** |
| Family support | | | 0.02 | -0.08 | 0.12 | 0.43 |
| Loneliness × Family support | | | -0.24** | -0.43 | -0.06 | -2.52** |
| **Conditional direct effect analysis at values of family support (*M ± SD*)** | | | | | | |
| | | | *B* | *Boot* SE | *Boot* LLCI | *Boot* ULCI |
| *M - 1SD* (2.73) | | | 0.45 | 0.10 | 0.25 | 0.65 |
| *M* (3.40) | | | 0.28 | 0.08 | 0.13 | 0.44 |
| *M + 1SD* (4.07) | | | 0.12 | 0.10 | -0.09 | 0.32 |
| **Conditional indirect effect analysis at values of family support (*M ± SD*)** | | | | | | |
| | | | *B* | *Boot* SE | *Boot* LLCI | *Boot* ULCI |
| *M - 1SD* (2.73) | | | 0.10 | 0.04 | 0.04 | 0.19 |
| *M* (3.40) | | | 0.06 | 0.03 | 0.02 | 0.13 |
| *M + 1SD* (4.07) | | | 0.03 | 0.03 | -0.03 | 0.09 |

Note. *N* = 582. Unstandardized regression coefficients are reported. Bootstrap sample size = 5000. *LL* = low limit, *CI* = confidence interval, *UL* = upper limit.

*$p < 0.05$

**$p < 0.01$

***$p < 0.001$.

mobile phone addiction and that this relation was mediated by anthropomorphism. Thus, Hypothesis 1 and 2 were supported.

In order to examine Hypothesis 3, two interaction effects were analyzed with PROCESS macro (Model 8) by Hayes [67]. There was a significant loneliness ×family support interaction effect on anthropomorphism ($B = -0.34$, $p < 0.05$) in mediator variable model. A significant loneliness × family support interaction effect on mobile phone addiction ($B = -0.24$, $p < 0.01$) in the dependent variable model. These findings indicated that both the association between loneliness and mobile phone addiction and the association between loneliness and anthropomorphism were moderated by family support.

Additionally, simple slope analyses were conducted to illustrate these significant interactions and explore whether slopes for the high-family support group (1 *SD* above the mean) were different from slopes for the low-family support group (1 *SD* below the mean) in the two

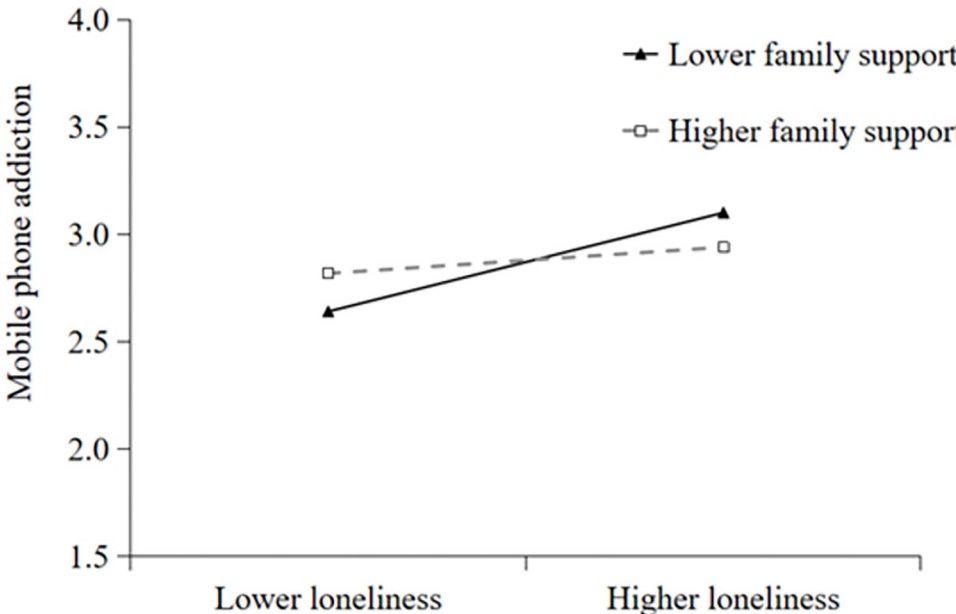

**Fig 2. Family support moderates the relation between loneliness and mobile phone addiction.**

models. The results were plotted in Figs 2 and 3. As shown in Fig 2, the effect of loneliness on mobile phone addiction was positive and significant for college students with low family support ($B = 0.55$, $t = 5.37$, $p < 0.001$), whereas it was not significant for those with high family support ($B = 0.14$, $t = 1.44$, $p > 0.05$). The results indicated that the direct effect of loneliness on mobile phone addiction was stronger for college students with lower family support. As shown in Fig 3, the effect of loneliness on anthropomorphism was positive and significant for college students with low family support ($B = 0.61$, $t = 3.20$, $p < 0.001$), whereas it was not

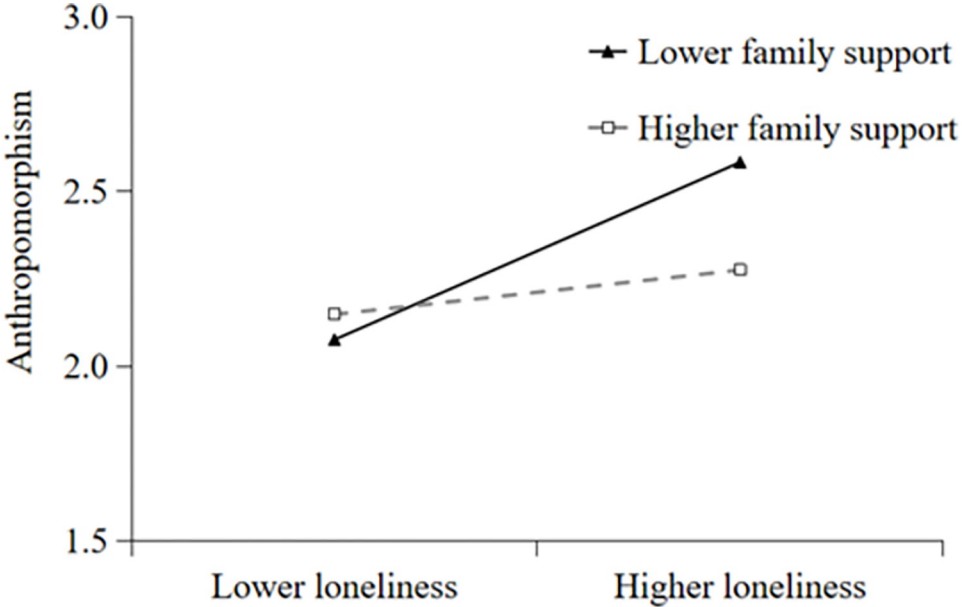

**Fig 3. Family support moderates the relation between loneliness and anthropomorphism.**

significant for those with high family support ($B = 0.15$, $t = 0.83$, $p > 0.05$). The results indicated that the indirect effect of anthropomorphism in the relationship between loneliness and mobile phone addiction was stronger for college students with lower family support.

In other words, the significant direct association between loneliness and mobile phone addiction, as well as the indirect effect of anthropomorphism in this relationship, was revealed among college students with low levels of family support but was observed to be non-significant among college students with high levels of family support, which indicates that as the level of family support decreases, both of these effects become stronger.

## Discussion

In the era of mobile Internet, mobile phones have even been regarded as a necessity part of the lives of college students [71]. With the increasing number of college students addicted to mobile phones, the potential risk factors of mobile phone addiction have attracted more and more researchers' attention. Researchers have conducted in-depth research into mobile phone addiction to better prevent and control it. Many studies have explored the risk factors closely related to mobile phone addiction, among which loneliness is one of the most concerning [11]. Correlation analyses indicated that loneliness was positively related to mobile phone addiction among Chinese college students. Hypothesis 1 was supported. Furthermore, the current study proposed a moderated mediation model to test the role of anthropomorphism and family support in the process. Considering the severe consequences of loneliness and the prevalence of mobile phone addiction, it is imperative to examine the mechanisms underlying the link between loneliness and mobile phone addiction. This study was designed to examine the relation between loneliness and mobile phone addiction and its underlying mechanisms among Chinese college students. Specifically, we proposed a moderated mediation model to analyze the role of anthropomorphism and family support in the relation between loneliness and mobile phone addiction. The moderated mediation analyses revealed that anthropomorphism mediated the association between loneliness and mobile phone addiction. Hypothesis 2 was therefore supported. The direct and indirect effects of loneliness on mobile phone addiction would be exacerbated when individuals cannot get support or satisfy their sense of belonging from their families in handling negative experiences induced by loneliness. In other words, college students with higher levels of family support could successfully alleviate the adverse effects of loneliness on mobile phone addiction. Hypothesis 3 was also supported. Findings revealed us that we could attenuate the potential adverse effects resulting from loneliness on our mental health by enhancing our family support.

First, consistent with a previous study conducted by Kim, Cho, and Kim [20], the present study indicated that loneliness could positively and significantly predict mobile phone addiction among college students. This finding confirmed that when people in the real world confronted with psychosocial problems (e.g., loneliness), will turn to the network or mobile phones to escape the pain [17]. When individuals with a high level of loneliness do not have enough social support, and their sense of belonging cannot be satisfied, they will be prompted to seek solutions to deal with negative psychological feelings [72]. To make up for the lack of social interaction in real life, individuals who suffer from loneliness are more inclined to seek social needs on the Internet [73]. The multiple functions of mobile phones can help individuals suffering from loneliness escape the painful psychological experience caused by unsatisfied needs in real life. For instance, the social function of mobile phones is beneficial for individuals suffering from loneliness in seeking online social support and gaining an online sense of belonging [12]. This finding illustrated that the use of mobile phones could satisfy various

psychological needs of individuals. Therefore, college students suffering from loneliness may have more mobile phone addiction.

Furthermore, this study also uncovered how loneliness is linked to mobile phone addiction by discovering that anthropomorphism is a critical mediating factor. In other words, loneliness could intensify individuals' anthropomorphism, which in turn promotes their risk of mobile phone addiction. This finding indicated that the negative emotions associated with loneliness lead to changes in individuals' emotional and cognitive responses to objects, further resulting in cognitive impairment and excessive attachment. Additionally, this study provided support for the cognition-behavior model [42]. Anthropomorphism is the proximal factor affecting the formation or maintenance of mobile phone addiction, while loneliness is the distal factor, and the distal factors (i.e., loneliness) influence mobile phone addiction through the mediating role of proximal factors (i.e., anthropomorphism). Individuals experience loneliness when they feel that their social relationships do not meet the criteria they expected [74]. Individuals suffering from loneliness make up for the lack of interpersonal relationships by establishing relationships with objects [29]. Anthropomorphic objects can help individuals form effective social and emotional connections [75], meeting their need to build social relationships [32]. Mobile phones represent a relationship-maintaining tool, making it much easier to become a compensatory attachment target than other objects [37]. In other words, individuals who suffer from loneliness feel a lack of acceptance and sense of belonging, leading them to use other ways (e.g., anthropomorphizing mobile phones as "companion", "secretary", "friend", or "safety ward") to meet their basic needs [40,76]. Besides, this study also provides support for the three-factor theory of anthropomorphism that individual anthropomorphism can reduce the negative emotions and externalize problem behaviors caused by loneliness [27]. Unfortunately, the excessive attachment to mobile phones following feelings of loneliness may persist if they excessively anthropomorphize their mobile phones rather than obtain social connections. Therefore, anthropomorphism induced by loneliness may lead to college students' excessive attachment to mobile phones, which will eventually evolve into mobile phone addiction Overall, anthropomorphism was an underlying mechanism for us to understand how loneliness influences mobile phone addiction.

In addition, one valuable finding of the present study was examination of individual differences in the predictive effects of loneliness on anthropomorphism and mobile phone addiction. Specifically, both the direct effect of loneliness on mobile phone addiction and the indirect effect via anthropomorphism were moderated and buffered by family support, with these effects being stronger for adolescents with lower levels of family support. These results indicated that family support, as one of the important positive external resources, does not only directly assuage mobile phone addiction but can also alleviate the potential adverse effects of loneliness on anthropomorphism and mobile phone addiction.

This finding is in line with the main points of resilience framework theory [52] and the risk-buffering model [58]. Both of these theories emphasize that family support can mitigate the adverse effects of risk factors. Individuals with positive external support resources (e.g., family support) are more adaptive when facing negative emotion. A possible explanation is that as an important positive external support, family resources provide individuals with physical, emotional, informational, and instrumental assistance perceived from their family members, which often act as an external coping resource to deal with negative emotion [54]. Previous research has confirmed that people in a good family atmosphere have a higher degree of engagement and satisfaction with the real world and life, and parental acceptance can promote individuals' positive self-evaluation [77]. Therefore, individuals with higher family support may be less likely to escape through their mobile phones or seek virtual social support when faced with loneliness [78]. Individuals who lack family support might feel isolated and

helpless when experiencing loneliness, and be more likely to seek virtual social support in online communication, which makes them more prone to mobile phone addiction. Additionally, the family support also affects one's cognitive adaptation. Anthropomorphism is argued to satisfy the individual's need to communicate with others [31]. Maintaining close relationships can reduce the individual's anthropomorphic tendencies [33]. As a part of an individual's intimate relationship, the family helps people to realize socialization, develop various abilities, create a favorable material and spiritual environment for their physical and psychological development, which can meet their development needs [79]. A high level of family support can provide individuals with intimate emotional connections to meet their basic psychological needs [80], weakening the impact of loneliness on anthropomorphism. Therefore, family support could buffer the priming effect of loneliness on anthropomorphism and mobile phone addiction. Unfortunately, individuals with lower family support cannot benefit from family members. The negative emotions of individuals with lower family support (e.g., parental rejection and neglect, harsh parenting) due to the lack of communication and interaction between individuals and family members cannot be relieved with the help of family members [60]. Individuals in a negative family environment who cannot change this situation will experience a sense of pressure and other negative emotions. Coping with these feelings will require some methods of relieving the negative emotions, such as anthropomorphism and mobile phone addition [81]. Furthermore, individuals with low family support have a low sense of psychological security and will use mobile phones to connect with and get support from others [13]. Therefore, individuals with lower family support are more likely to be involved in anthropomorphism and mobile phone addiction when they suffer from loneliness.

## Implications

In addition, these findings provide empirical evidence for formulating programs aimed at reducing the negative influences of loneliness, as well as for preventing and managing individuals with mobile phone addiction.

First, individuals might take preventive measures to reduce negative emotions such as loneliness and helplessness. Empirical evidence has suggested that interpersonal skills can predict individuals' loneliness [82]. Therefore, educators should take measures to train and improve individual social skills, provide social support resources, and implement cognition-related interventions to help individuals reduce loneliness. From the educational authority level, educational funding input can be increased, colorful courses and activities set up, and the sense of involvement and experience in individuals at school can be increased. From the teacher level, we can actively encourage and demonstrate, and guide individuals to participate in school activities and peer activities. At the same time, enhancing home-school cooperation, such as schools popularize to parent rational ways and right knowledge to reduce loneliness and mobile phone addiction.

Second, given that anthropomorphism as a "bridge" mediates the relationship between loneliness and mobile phone addiction, it would be of great benefit if individuals could reduce their tendency to anthropomorphize their mobile phones. According to the self-determination theory, if the social environment cannot meet individuals' psychological needs for a long time, they will have a strong desire to satisfy these needs from other social environments, such as the virtual world of mobile phones [83]. One of the reasons why individuals tend towards anthropomorphism is to satisfy their motivation for sociability [27]. Therefore, parents and educators should pay attention to cultivating adolescents' social communication skills. For example, educators teach students interpersonal skills, and parents encourage their children to engage in peer interactions and to engage in hands-on activities, as a way to increase individuals'

investment and confidence in their social competence and thus avoid falling alone and mobile phone addiction.

Finally, the results of this study also indicated that for individuals with high family support, the association between loneliness and mobile phone addiction is not significant. Thus, family support is a complementary component of intervention programs aimed at decreasing mobile phone addiction, such as providing emotional comfort, instrumental aid, and conveying understanding, and acceptance [84]. Parents currently display external violent aggression in the process of raising children, such as yelling or beating their children [85] as well as internal emotional attitudes, such as ignoring, opposing, refusing, and threatening their children [86]. According to parental acceptance-rejection theory, parents' behaviors such as rejection can make the internet more attractive to adolescents [87], which makes people more prone to mobile phone addiction [88]. Therefore, parents should provide their children with a democratic parenting style, and increase the communication with children and the perception of parent-child relationship intimacy as methods to reduce the likelihood of mobile phone addiction.

## Limitations of the study

Although this study provides valuable findings for understanding how and when loneliness influences individuals' anthropomorphism and mobile phone addiction, the current study is not without limitations. First, this was a cross-sectional study, which means that the examination of causality between loneliness and individuals' mobile phone addiction could not be examined. Future studies should consider a longitudinal design to get a more persuasive conclusion of the relationship between loneliness, anthropomorphism, family support, and mobile phone addiction. Then, the current study demonstrated that anthropomorphism is not only a possible adverse consequence of loneliness but also closely related to individuals' mobile phone addiction. Notably, anthropomorphism only partially mediated the relationship between loneliness and mobile phone addiction. Thus, other factors (e.g., psychological security [13]) should be considered. Moreover, as a supportive but non-directive relationship, family support might not provide critical buffering resources (e.g., behavior monitoring and emotion regulation skills) to prevent mobile phone addiction for adolescents with high levels of loneliness [89]. Considering these reasons, future research must explore the moderating role of other important protective factors (e.g., parent-child communication and teacher-student relationship) in the relationship between loneliness and mobile phone addiction.

## Conclusions

To sum up, the present study enriches our understanding of the mechanisms linking loneliness to mobile phone addiction: anthropomorphism was shown to be a partial mediator of the link between loneliness and mobile phone addiction. Furthermore, family support was shown to exert a positive moderating effect on this association. Family support could especially alleviate the direct influence of loneliness on mobile phone addiction and the indirect effect through the mediating effect of anthropomorphism. The mediating role of anthropomorphism and the moderating role of family support together contribute to uncovering an answer to how loneliness is associated with mobile phone addiction and when this association is more pronounced or weaker. At the same time, these findings also expand the role of family support as one of the protective factors to buffer against the deleterious influences of negative emotions such as loneliness.

## Supporting information

**S1 Data.**
(SAV)

## Acknowledgments

We are grateful to the school administrators, teachers and college students who actively cooperated with the researchers to collect data.

## Author Contributions

**Conceptualization:** Yanhong Zhang, Yongxue Li, Lei Yan, Shuailei Lian.

**Data curation:** Yanhong Zhang, Lei Yan, Shuailei Lian.

**Formal analysis:** Yanhong Zhang, Yongxue Li, Meng Xia.

**Methodology:** Yongxue Li, Meng Xia, Miao Han.

**Writing – original draft:** Yanhong Zhang, Yongxue Li, Lei Yan, Shuailei Lian.

**Writing – review & editing:** Yanhong Zhang, Yongxue Li, Miao Han.

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
