## [Decision Letter · Decision Letter 0]

25 Oct 2022

PONE-D-22-21299Loneliness and mobile phone addiction among Chinese college students: the mediating role of anthropomorphism and moderating role of family supportPLOS ONE

Dear Dr. Lian,

Thank you for submitting your manuscript to PLOS ONE. After careful consideration, we feel that it has merit but does not fully meet PLOS ONE’s publication criteria as it currently stands. Therefore, we invite you to submit a revised version of the manuscript that addresses the points raised during the review process.

We look forward to receiving your revised manuscript.

Kind regards,

Arista Lahiri

Academic Editor

PLOS ONE

Journal Requirements:

2. PLOS ONE does not copy edit accepted manuscripts (https://journals.plos.org/plosone/s/criteria-for-publication#loc-5). To that effect, please ensure that your submission is free of typos and grammatical errors. 

3. Please change "female” or "male" to "woman” or "man" as appropriate, when used as a noun (see for instance https://apastyle.apa.org/style-grammar-guidelines/bias-free-language/gender)

5. We note that you have stated that you will provide repository information for your data at acceptance. Should your manuscript be accepted for publication, we will hold it until you provide the relevant accession numbers or DOIs necessary to access your data. If you wish to make changes to your Data Availability statement, please describe these changes in your cover letter and we will update your Data Availability statement to reflect the information you provide

Additional Editor Comments (if provided):

Please address the concerns raised by the reviewers

Reviewers' comments:

Reviewer's Responses to Questions

**Comments to the Author**

1. Is the manuscript technically sound, and do the data support the conclusions?

Reviewer #1: Partly

Reviewer #2: Yes

2. Has the statistical analysis been performed appropriately and rigorously? 

Reviewer #1: Yes

Reviewer #2: Yes

3. Have the authors made all data underlying the findings in their manuscript fully available?

Reviewer #1: Yes

Reviewer #2: No

4. Is the manuscript presented in an intelligible fashion and written in standard English?

Reviewer #1: Yes

Reviewer #2: Yes

5. Review Comments to the Author

Reviewer #1: The manuscript is well written but there are some issues which needs clarification:

1. Some references could not be found. Line 214, 216

2. What is the rationale of considering participants of 17-24 age groups in the study and mostly "adolescents" has been mentioned in the manuscript, what according to the authors is considered as adolescent age group?

3. The total number of participants mentioned in the study is 582.But when categorising, the total no.of participants is not coming as 582. Line 248-251

4. Give proper explanation about rationality of using Harman Single factor test with appropriate references.

5. Under Limitation section line 487,488: Regarding common method biases, this limitation has been mitigated. So better to avoid mentioning under limitation section and can be discussed in Discussion section of the study.

6.Role of teachers and school authorities to be elaborated more in preventing loneliness and mobile addiction under Conclusion section.

7. How the sample size was calculated is not clear.

Reviewer #2: Title: Looks Incomplete

Abstract: Better if written in structural form

Introduction: Very much elaborated, can be precise and related to objectives. The explanation on mediating role and moderating role needs to be better clarified.

Materials and methods:

• Study type and design-Needs to be clearly mentioned.

• Sample size calculation-was there any basis of the sample size calculation?

• Inclusion criteria-Needs to be clearly mentioned

• Pre testing- Was any pre testing conducted?

• Ethical approvals-Was the ethical approval taken for the study?

Conclusion and recommendation: Need to be separated

6. PLOS authors have the option to publish the peer review history of their article (what does this mean?). If published, this will include your full peer review and any attached files.

Reviewer #1: No

Reviewer #2: No

---

## [Author Response · Author response to Decision Letter 0]

5 Feb 2023

Reviewer #1: The manuscript is well written but there are some issues which needs clarification:

1. Some references could not be found. Line 214, 216

R: Thank the experts on reminding. Due to our mistakes, the citation format of these two references has been omitted and has been re added in the manuscript.

2. What is the rationale of considering participants of 17-24 age groups in the study and mostly "adolescents" has been mentioned in the manuscript, what according to the authors is considered as adolescent age group?

R: Thanks for expert suggestion. We consulted relevant literature and books and found no uniform standards for different academics and organizations regarding the chronological age stage division of adolescents. The current "adolescents" age definition is 12,13-17,18 years; 13,14-28years; Had 7,8-28years; 7,8-40 years; And 13,14-40 years et al. 

Combined with the practical need for investigation into this study, we finally consider the 17-24 years age group participants for two main reasons. First according to Zhang's《psychology of adolescent development》, the term adolescence should refer to the social group defined by the terms juvenile (11,12-14,15) , early youth(14,15-17,18), and late youth(17,18-24,25) , which means 11,12-24,25 years of age. The college student group was in advanced youth stages. Second, due to the limitation of time and economic cost, considering the effectiveness and economy of data distribution and recovery, we adopted a convenient sampling method to collect data by issuing online questionnaires to college students. Because college students have more opportunities and time to use mobile phones independently than junior high school students and senior high school students, which is helpful for online data collection.

Based on the physical and psychological growth characteristics, we think that the adolescent age group refers to the group 11,12-23,24 which contains the early youth, middle youth and late youth groups, and the college student group is in the late youth group. In addition, our review of the literature found that Gao et al (2023), Sugimura et al (2022) articles also counted college student samples in the adolescent group.

Attached references:

Zhang, WX. Adolescent developmental psychology. Jinan: Shandong people's press, 2008.

Gao, F., Bai, XJ., Zhang, P., Cao, HB. A Meta-analysis of the Relationship between Parenting Styles and Suicidal Ideation in Chinese Adolescents. Psychological Development and Education.2023;(01):97-108.https://doi.org/10.16187/j.cnki.issn1001-4918.2023.01.11.

Sugimura, K., Hihara, S., Hatano, K.et al. Profiles of Emotional Separation and Parental Trust from Adolescence to Emerging Adulthood: Age Differences and Associations with Identity and Life Satisfaction. J Youth Adolescence. 2022; Dec 16. https://doi.org/10.1007/s10964-022-01716-z PMID:36525106

3. The total number of participants mentioned in the study is 582.But when categorising, the total no.of participants is not coming as 582. Line 248-251

R: Thanks for expert suggestion. Our grade description of the manuscript lacks senior year, and the data has been added to the manuscript after re verification. See lines 251-253.

The number of students in each grade is as follows:

One hundred and twenty-two (20.96%) of them were freshmen; one hundred and forty-two (24.40%) of them were sophomores; one hundred and seventy-nine (30.76%) of them were juniors; One hundred and thirty-nine (23.88%) of them were seniors.

4. Give proper explanation about rationality of using Harman Single factor test with appropriate references.

R: Thanks for expert suggestion.

Harman single factor test, the basic assumption of this technique is that if there is a large number of method variations, a single factor can be separated during factor analysis; Either a common factor explains most of the variation in variables. The biggest advantage of Harman single factor test is that it is simple and easy to use, but it is only a diagnostic technique to evaluate the variation severity of common methods, and it is also an insensitive test method, without any role in controlling the effect of methods. According to its assumption, only when a single factor separates from the factor analysis and explains most of the variation in variables, it is reasonable to think that there is a serious common method deviation.

In addition, we reviewed the relevant literature and found that although Harman Single factor test was widely used in a large number of studies, it has also been questioned by some scholars in recent years. For example, Schwarz et al (2017) and Miguel I. Aguirre Ureta & Jiang Hu (2019) believed that it was worthless to continue to rely on Harman's single factor test to provide evidence against substantive methodological bias in empirical research, call for abandoning this method and looking for more effective alternatives.

Although Harman single factor test is still widely used, of course, we also expect researchers to find effective alternative methods as soon as possible to help you carry out research.

References attached:

Podsakoff PM, MacKenzie SB, Lee JY, Podsakoff NP. Common method biases in behavioral research: a critical review of the literature and recommended remedies. J Appl Psychol. 2003; 88:879–903. https://doi.org/10.1037/0021-9010.88.5.879 PMID:14516251

Zhou H, Long LR. Statistical remedies for common method biases. Adv Psychol Sci. 2004;12(6):942-50.

Schwarz, A., Rizzuto, T., Carraher-Wolverton, C., Roldan, J., & Barrera, R. Examining the impact and detection of the “urban legend” of common method bias. The DATA BASE for Advances in Information Systems, 2017;48(1).

Aguirre-Urreta MI, Hu J. Detecting common method bias: performance of the Harman's single-factor test. ACM SIGMIS Database. 2019; 50:45–70. https://doi.org/10.1145/3330472.3330477

5. Under Limitation section line 487,488: Regarding common method biases, this limitation has been mitigated. So better to avoid mentioning under limitation section and can be discussed in Discussion section of the study.

R: Thanks for expert suggestion. We delete this content with the limitation section of the manuscript.

6.Role of teachers and school authorities to be elaborated more in preventing loneliness and mobile addiction under Conclusion section.

R: Thanks for expert suggestion. We adapted the conclusion section in the manuscript to differentiate it into conclusion and implications sections. And further describe the role of teachers and school authorities on preventing loneliness and mobile phone addiction in the implications section. See lines 485-491 and 499-502.

7. How the sample size was calculated is not clear.

R: Thanks for expert suggestion. In this study, we used G*Power3.1.9.7 to calculate the sample size, the calculated parameters including Tails = two, Effect size = 0.2, α err prob = 0.05, Power (1-β) = 0.95, calculated sample size is 314. Considering the invalid response rate of the subjects, assuming that the invalid response rate is 20%, 314 / (1-0.2) = 393 questionnaires should be sent out. We have supplemented this to the participants and procedures section of the manuscript, see lines236-239.

References attached:

Kang H. Sample size determination and power analysis using the G*Power software. J Educ Eval Health Prof. 2021;18.17. https://doi.org/10.3352/jeehp.2021.18.17

Reviewer #2: 

Title: Looks Incomplete

R: Thanks for expert suggestion. We revised the title of the article as follows:

The relationship between loneliness and mobile phone addiction among Chinese college students: the mediating role of anthropomorphism and moderating role of family support.

Abstract: Better if written in structural form

R: Thanks for expert suggestion. We adapted the abstract section in the manuscript in structured form to make it appear more rational and legible.

Introduction: Very much elaborated, can be precise and related to objectives. The explanation on mediating role and moderating role needs to be better clarified.

R: Thanks for expert suggestion. We have revised the introduction and cut some of the contents to make it closer to the research purpose. At the same time, we also further to explain the mediation and moderation functions to make them clearer. See the introduction section of the manuscript.

Materials and methods:

• Study type and design-Needs to be clearly mentioned.

R: Thanks for expert suggestion. This study constructs a moderated mediational research model by reviewing literature and proposing hypotheses. Research on the association with loneliness with cell phone addiction and its underlying mechanisms. We have adapted the description of this study design in our manuscript to make it appear clearer and unambiguous.

• Sample size calculation-was there any basis of the sample size calculation?

R: Thanks for expert suggestion. In this study, we used G*Power3.1.9.7 to calculate the sample size, the calculated parameters including Tails = two, Effect size = 0.2, α err prob = 0.05, Power (1-β) = 0.95, calculated sample size is 314. Considering the invalid response rate of the subjects, assuming that the invalid response rate is 20%, 314 / (1-0.2) = 393 questionnaires should be sent out. We have supplemented this to the participants and procedures section of the manuscript, see lines 236-239.

References attached:

Kang H. Sample size determination and power analysis using the G*Power software. J Educ Eval Health Prof. 2021;18.17. https://doi.org/10.3352/jeehp.2021.18.17

• Inclusion criteria -Needs to be clearly mentioned

R: Thanks for expert suggestion. Inclusion criteria for this study were full-time college students, filling in longer than 3 mins, and volunteering to participate in this survey. We supplement this to the manuscript, see lines 242-243.

• Pre testing - Was any pre testing conducted?

R: Thanks for expert suggestion. There was no pre-test in this study. Pre-test is often used in the research of questionnaire preparation and experiment to design. In this study, we used a mature questionnaire with good reliability and validity to collect data. At the same time, we also consulted the relevant literature, and found that the researchers did not conduct a pre-test when using the same type of questionnaires for research, so this study did not conduct a pre-test. However, we will improve on this in future study.

• Ethical approvals-Was the ethical approval taken for the study?

R: Thanks for expert suggestion. In any research, ethics is necessary. We also attach great importance of ethical principles. This study strictly abides by the ethical standards of psychology and the ethical research standards of our university. We applied to the psychology department of our university for ethical approval and were approved.

Conclusion and recommendation: Need to be separated

R: Thanks for expert suggestion. We readjusted this part of the manuscript, separating conclusions and recommendations to make them clearer and more logical.

---

## [Editor Report · Decision Letter 1]

18 Apr 2023

The relationship between loneliness and mobile phone addiction among Chinese college students: the mediating role of anthropomorphism and moderating role of family support

PONE-D-22-21299R1

Dear Dr. Lian, 

We’re pleased to inform you that your manuscript has been judged scientifically suitable for publication and will be formally accepted for publication once it meets all outstanding technical requirements.

Kind regards,

Yaser Mohammed Al-Worafi

Academic Editor

PLOS ONE
---

## [Editor Report · Acceptance letter]

20 Apr 2023

PONE-D-22-21299R1 

The relationship between loneliness and mobile phone addiction among Chinese college students: the mediating role of anthropomorphism and moderating role of family support 

Dear Dr. Lian:

I'm pleased to inform you that your manuscript has been deemed suitable for publication in PLOS ONE. Congratulations! Your manuscript is now with our production department. 

Kind regards, 

on behalf of

Professor Yaser Mohammed Al-Worafi 

Academic Editor

PLOS ONE